# Prevalence of Zinc Deficiency in Healthy 1–3-Year-Old Children from Three Western European Countries

**DOI:** 10.3390/nu13113713

**Published:** 2021-10-22

**Authors:** Mirjam Vreugdenhil, Marjolijn D. Akkermans, Liandré F. van der Merwe, Ruurd M. van Elburg, Johannes B. van Goudoever, Frank Brus

**Affiliations:** 1Department of Pediatrics, Juliana Children’s Hospital, Haga Teaching Hospital, 2545 AA The Hague, The Netherlands; m.d.akkermans@hagaziekenhuis.nl (M.D.A.); fbrus1957@gmail.com (F.B.); 2Danone Nutricia Research, 3584 CT Utrecht, The Netherlands; liandre.vandermerwe@danone.com; 3Department of Pediatrics, Amsterdam UMC, Vrije Universiteit, University of Amsterdam Emma Children’s Hospital, 1105 AZ Amsterdam, The Netherlands; rm.vanelburg@amsterdamumc.nl (R.M.v.E.); h.vangoudoever@amsterdamumc.nl (J.B.v.G.); 4Nutriton4Health, 1214 LT Hilversum, The Netherlands

**Keywords:** zinc, zinc deficiency, dietary zinc intake, children, Western Europe

## Abstract

Zinc deficiency (ZnD) has adverse health consequences such as stunted growth. Since young children have an increased risk of developing ZnD, it is important to determine its prevalence and associated factors in this population. However, only a few studies have reported on ZnD prevalence in young children from Western high-income countries. This study evaluated ZnD prevalence and associated factors, including dietary Zn intake, in healthy 1–3-year-old children from Western European, high-income countries. ZnD was defined as serum Zn concentration <9.9 µmol/L. A total of 278 children were included with a median age of 1.7 years (Q1–Q3: 1.2–2.3). The median Zn concentration was 11.0 µmol/L (Q1–Q3: 9.0–12.2), and ZnD prevalence was 31.3%. No significant differences were observed in the socio-economic characteristics between children with and without ZnD. Dietary Zn intake was not associated with ZnD. ZnD is common in healthy 1–3-year-old children from Western European countries. However, the use of currently available cut-off values defining ZnD in young children has its limitations since these are largely based on reference values in older children. Moreover, these values were not evaluated in relation to health consequences, warranting further research.

## 1. Introduction

Zinc is an important trace element for human health, as it is involved in numerous and diverse biological processes throughout the body, including cell growth and differentiation, gene expression, and protein synthesis [1,2,3]. Subsequently, zinc deficiency (ZnD) has a wide range of possible adverse health consequences, including stunted growth, skin lesions and delayed wound healing, and impaired innate and adaptive immune functions [4,5,6,7,8].

ZnD is highly prevalent worldwide, resulting in substantial disease burden, especially in young children since they have an increased risk of developing ZnD due to increased requirements during periods of rapid growth [9,10]. In low- and middle-income countries, varying prevalence rates of ZnD have been reported in infants, young children, and preschool-age children, ranging from 5.1% in Sri Lanka to 82.6% in Cameroon [11]. Only a few studies report on the prevalence rates of ZnD in healthy young children from Western high-income countries, ranging from 0–60% [12,13,14,15,16,17]. However, most of these studies were conducted some time ago, while dietary habits in Western countries may have changed over the years. Moreover, in most of these studies, study populations were relatively small, and associated factors were not extensively assessed.

Because of the possible adverse health consequences of ZnD in young children, it is important to determine ZnD prevalence and associated factors in this population to implement and guide preventive strategies. Therefore, this study aimed to assess the prevalence of ZnD in a population of healthy 1–3-year-old children living in Western Europe and evaluate the possible associated factors, including dietary Zn intake.

## 2. Materials and Methods

### 2.1. Study Design

Data were used from a previously conducted randomized, double-blind controlled trial (NTR3609) investigating the effect of a micronutrient-fortified growing-up milk given for 20 weeks on the iron and vitamin D status of healthy 1–3-year-old children in comparison to non-fortified cow’s milk [18]. This study was conducted from October 2012 to September 2014 in three Western European countries, namely Germany (nine private pediatric clinics), the Netherlands (one secondary and two tertiary hospitals), and the United Kingdom (UK) (one secondary and one tertiary hospital). The study protocol was approved by the medical ethical review boards of all participating sites and has been previously described in detail [18]. In addition to iron and vitamin D status, other micronutrients, including zinc (Zn), were also analyzed. Blood samples collected during the baseline study visit were used for the current study.

### 2.2. Study Population and Procedure

Children between 1 and 3 years of age with a stable health status (i.e., no known chronic or recent acute diseases, no known infection during the last week, or infection needing medical assistance or treatment during the last two weeks) and expected to remain stable were eligible for participation. The recruitment of participants differed between countries. In the Netherlands and the UK, parent(s) or legal representative(s) of eligible children were informed about the study during a preoperative visit before an elective non-emergency surgical procedure (e.g., urologic surgeries, inguinal or umbilical hernia operations, or ear–nose–throat procedures). If written informed consent was obtained from the parent(s)/legal representative(s), the baseline study visit coincided with the elective non-emergency surgical procedure. The baseline blood sample was then combined with intravenous injection necessary for administering general anesthesia. In Germany, participants were recruited during a regular visit to a pediatric clinic. During the baseline study visit, the child’s body weight and height/length were measured. Body weight was measured while wearing only underwear, using a calibrated weighing scale. Height was measured while standing and without wearing shoes, using a calibrated stadiometer. Length was measured in children for whom it was not possible to take a height measurement while standing, using a length board. Weight-for-age-z-scores and height/length-for-age-z-scores were calculated using World Health Organization (WHO) Child Growth Standards. Stunting was defined as the height/length-for-age >2 standard deviations (SD) below the median of WHO Child Growth Standards [19]. Moreover, parent(s)/legal representative(s) were asked questions during the baseline visit about their child’s demographic- and socio-economic characteristics, daycare center attendance, medical history, and dietary intake [18]. Dietary intake was estimated using a food-frequency questionnaire adapted from other dietary questionnaires [20,21], as previously described [18]. This questionnaire reflects dietary intake during the month before the baseline visit. For this study, we mainly focused on the food groups known to affect zinc status (e.g., meat, vegetables, fish, vegetables, grains and grain-based products, and eggs). Zn intake (mg/day) (per reported food group) was determined by using the Dutch Food Composition Database [22].

### 2.3. Laboratory Analysis and Definitions

Venous blood samples were collected from participants throughout various times of the day by trained personnel using trace-element free heparinized tubes, following local laboratory protocol and procedures recommended by the International Zinc Nutrition Consultative Group (IZiNCG) [19]. After clotting and centrifugation, serum was distributed and aliquoted in polypropylene tubes and stored between −20 °C and −80 °C at the study site. From the study sites, samples were shipped (within two months) on dry ice to the Nutricia Research Analytical Science Laboratory in Utrecht, the Netherlands, where samples were stored at −80 °C. From the Nutricia Research Analytical Science Laboratory, samples were shipped on dry ice to the Reinier Haga Medical Diagnostic Center in Delft, the Netherlands, where samples were stored at −80 °C until analysis. Serum Zn concentrations were determined using flame atomic absorption spectrophotometry (AA-7000, Shimadzu) and hsCRP concentrations using a clinical chemistry analyzer (Abbott Architect c16000) with a turbidimetry method. ZnD was defined as a serum Zn concentration <9.9 µmol/L [19]. Since Zn concentration may be reduced in the case of infection or inflammation [23,24,25], participants with a high-sensitivity C-reactive protein (hsCRP) concentration >5 mg/L [26] were excluded from analyses. Additionally, children with missing Zn and/or hsCRP measurements at baseline were also excluded from analyses.

### 2.4. Statistical Analysis

SPSS version 24.0 (SPSS Inc., Chicago, IL, USA) was used for all analyses. Distribution of data was assessed using histograms and Shapiro–Wilk tests. Data are presented as the mean with SD for normally distributed variables or median with first and third quartiles (Q1–Q3) for non-normally distributed variables. Categorical variables are presented as numbers with percentages. Comparisons between groups were made using the independent T-test for normally distributed variables and the Mann- Whitney U non-parametric test for non-normally distributed variables. For categorical variables, comparisons between groups were made using the Chi-squared or Fisher’s exact test. Statistical significance was defined as *p* < 0.05. In order to further explore factors associated with ZnD, a binary logistic regression analysis was performed using the variables previously identified with a *p*-value < 0.1 as covariates, including age and sex independent of *p*-values, and ZnD as dependent variable.

## 3. Results

### 3.1. Study Population

In total, 278 of the 325 (85.5%) healthy 1–3-year-old children of the original nutritional intervention study were included for our Zn analyses: 237 (85.3%) in Germany, 37 (13.3%) in the Netherlands, and 4 (1.4%) in the UK. Children excluded from analyses had missing Zn and/or hsCRP measurements or an hsCRP concentration >5 mg/L at baseline. There were no statistically significant differences regarding socio-economic characteristics and dietary intake found between included and excluded children (data not shown). The median age of the study participants was 1.7 years (1.2–2.3). Of the study population, 153 children (55.0%) were male, and 267 (96.0%) were Caucasian. Stunting was present in three children (1.1%). More characteristics of the study population are shown in Table 1.

### 3.2. Serum Zn Concentrations, ZnD Prevalence and Dietary Zn Intake

In the total study population, the median Zn concentration was 11.0 µmol/L (9.0–12.2). The median Zn concentrations in children with and without ZnD were 9.0 µmol/L (8.0–9.0) and 12.0 µmol/L (10.7–13.0), respectively. ZnD was present in 87 (31.3%) children. The median total dietary Zn intake (from milk and solid foods) was 5.71 mg/day (4.43–6.85), with a median Zn intake from milk of 2.78 mg/day (1.95–3.57) and from solid foods of 2.52 mg/day (1.86–3.73). More dietary intake details of the study population are shown in Table 1.

### 3.3. Factors Associated with ZnD

No significant differences in sex distribution, ethnicity, parental educational or professional status, or daycare attendance were found between children with and without ZnD. Children with ZnD were significantly more often ‘from Germany’ (98.9%) compared to children without ZnD (79.1%, *p* < 0.001). Furthermore, no significant differences in the consumption of solid foods (gr/day) or Zn intake (mg/day) per reported food group in children with and without ZnD were found. Finally, no significant differences in Zn intake from milk (i.e., formula, cow’s milk, or otherwise) (*p* = 0.481), solid foods (*p* = 0.066), or total dietary Zn intake (from milk and solid foods) (*p* = 0.978) between children with and without ZnD were observed (Table 1).

A binary logistic regression analysis was performed to further explore factors associated with ZnD, with ‘ZnD’ as dependent variable and ‘age’, ‘sex’, ‘ever breastfed’, ‘from Germany’, and ‘dietary Zn intake’ as covariates, which showed that being ‘from Germany’ was significantly associated with ZnD (OR = 16.6; 95% CI 92.2–124.2; *p* = 0.006). No other significant associations with ZnD were found (data not shown). Although ‘Zn intake from dried fruits, seeds and nuts’ had a *p*-value < 0.1, we chose not to include this variable in our multivariate model since this variable was already included in ‘dietary Zn intake’. A separate binary logistic analysis with ‘Zn intake from dried fruits, seeds and nuts’ instead of ‘dietary Zn intake’, showed no significant association with ZnD (data not shown).

A subgroup analysis was performed in 1–2-year-old and 2–3-year-old children: ZnD was present in 54 (30.2%) and 33 (33.3%) of 1–2-year and 2–3-year-old children, respectively, and this difference in ZnD prevalence was not significant (*p* = 0.586).

## 4. Discussion

This study showed that ZnD, with a prevalence rate of 31.3%, is common in healthy 1–3-year-old children living in three Western European countries. In addition to being from Germany, no other evaluated factors, including total dietary Zn intake, were associated with ZnD.

### 4.1. ZnD Prevalence

With an overall prevalence of 31%, we showed that ZnD is not uncommon in healthy 1–3-year-old children living in three Western European high-income countries. In low- and middle-income countries, prevalence rates of ZnD between 5% and 83% have been reported in young children [11]. Populations in low- and middle-income countries are at an increased risk of inadequate Zn intake and subsequently ZnD, which can be partially attributed to limited access to foods that are rich in zinc, such as animal products, in combination with a mainly plant-based diet, which contains phytates that inhibit intestinal Zn absorption [27]. Studies conducted in young children in Western high-income countries also report varying prevalence rates of ZnD, ranging between 0% and 60% in 1-year-old children [12,14,15,17]. ZnD was found in 21% of healthy children below three years of age in France [13] and in 38% of healthy toddlers between 12 and 20 months of age in New Zealand [16]. However, the comparison of prevalence rates of ZnD between studies remains challenging due to relatively small sample sizes and differences in study populations.

### 4.2. Factors Associated with ZnD

In our study population, we found no association between ZnD and the evaluated sociodemographic factors, besides being from Germany. Children with ZnD were significantly more often ‘from Germany’ (98.9%) compared to children without ZnD (79.1%). We cannot rule out that this association is possibly due to differences in setting and recruitment procedures (e.g., fasting state) between children recruited from Germany and children recruited from the Netherlands and the UK. However, we believe that since the majority of participants were from Germany (85%), it is likely by chance that children with ZnD are more likely to come from Germany than from the Netherlands or the UK. Moreover, no association between ZnD and dietary Zn intake was found. Three meta-analyses also concluded that serum Zn concentrations are unrelated to Zn intake [28,29,30]. Furthermore, Hennigar et al. [31] reported no differences in serum Zn concentrations between those with lower or higher Zn intake than the recommended dietary allowance for Zn. With a median total dietary Zn intake of 5.6 mg/day, our study population meets the recommended Zn intakes in each participating country [32,33,34]. The lack of association between dietary Zn intake and Zn concentration might be explained by factors that influence intestinal Zn absorption [35,36,37]. As mentioned before, phytates inhibit Zn absorption [27], whereas an increased amount of dietary protein, for example, increases Zn absorption [38,39]. The type of dietary protein also affects Zn absorption, for example, animal-derived proteins increase Zn absorption compared to plant-derived proteins [40,41]. In short, the composition of the diet might impact intestinal Zn absorption, despite sufficient Zn intake, and this might (partially) explain why we found no association between ZnD and dietary Zn intake. 

### 4.3. Strengths and Limitations

The strengths of this study are the relatively large study population of healthy 1–3-year-old, mainly Caucasian, children from three Western European countries, and the effect of inflammation and/or infection on Zn concentrations that was taken into account by excluding children with CRP > 5 mg/L. It is known that Zn concentrations can be affected by factors such as inflammation, fasting state, and/or diurnal rhythm [42]. A limitation of this study is that we did not have information concerning the fasting state during blood draw, the time of blood draw, and meal consumption prior to the blood draw, which might have influenced the reported Zn concentrations [31]. The cut-off value of 9.9 µmol/L (65 µg/dL) used to define ZnD, as suggested by the IZiNCG, accounts for blood samples collected in the morning with a non-fasting state. For blood samples collected in the afternoon, the IZiNCG suggests a cut-off value of 8.7 µmol/L (57 µg/dL) [19]. Since we do not know the exact time the blood samples were collected, we chose the highest cut-off value for a non-fasting state (i.e., 9.9 µmol/L) to define ZnD, which may have resulted in an overestimation of the prevalence of ZnD in our study population. However, when using the cut-off value of 57 µg/dL (8.7 µmol/L) for afternoon samples, we found the prevalence of ZnD to be 11.5%, which is still relatively high.

### 4.4. Defining ZnD

Finally, we want to address some concerns regarding the current definition of ZnD. At present, Zn concentration in either plasma or serum is the most extensively investigated and widely used biomarker to assess zinc status. Currently, the cut-off values for determining ZnD are largely based on reference values derived from large cohorts of presumably healthy people, such as the second National Health and Nutrition Examination Survey (NHANES II) during the period 1976–1980 [43]. Based on the results from the NHANES II survey, cut-off values for ZnD are suggested by the IZiNCG, i.e., <65 µg/dL (9.9 µmol/L) in the morning, and <57 µg/dL (8.7 µmol/L) in the afternoon for children below ten years of age [19], and are widely used in most studies evaluating zinc status in humans. However, children under the age of three were not represented in this survey. It is, therefore, possible that these cut-off values do not apply for 1–3-year-old children. Two studies evaluated serum Zn concentrations in healthy 1–3-year-old children living in Australia and Belgium [44,45]. First, reference intervals (2.5th and 97.5th percentiles) for serum Zn concentrations reported by Karr et al. were 9–19 µmol/L and 8–19 µmol/L in 1–2-year-old and 2–3-year-old healthy Australian children [44]. Second, van Biervliet et al. found mean serum Zn concentrations of 12.7 µmol/L (SD 2.8 µmol/L) in 1–2-year-old children and 12.7 µmol/L (SD 3.1 µmol/L) in 2–3-year-old healthy Belgian children [45,46]. These values do not differ that much from those resulting from the NHANES II survey in 3–4-year-old children (i.e., 12.2 µmol/L (SD 2.0 µmol/L)) [46]. However, both studies did not clearly take into account the presence of infection or the time of blood sampling which both can influence Zn concentration [44,45]. 

However, Zn concentration might not accurately reflect zinc status. Zn is present in all tissues, of which skeletal muscle and bone have the highest zinc content (approximately 83% of total body zinc), whereas plasma accounts for less than 0.2% of total body zinc. It is currently unknown whether Zn concentrations below a certain predefined cut-off value (e.g., based on cohort-derived reference values) fail to meet bodily requirements and cause negative health consequences. In adults, a lower cut-off value of 50 µg/dL (i.e., 7.6 µmol/L) has been proposed for clinical ZnD associated with symptoms, including diarrhea and alopecia [47]. To our knowledge, no such studies have been conducted in (young) children (to date). In our study, stunting (i.e., a functional health outcome of ZnD) was only prevalent in three children (1.1%), and due to this small number, no statistical analysis was performed regarding a possible association with ZnD. Furthermore, other biomarkers for assessing Zn status have been evaluated. For example, urinary zinc excretion, hair zinc concentration [48], or zinc-binding proteins, such as metallothionein [42], might be promising biomarkers for assessing human zinc status. However, due to a limited number of studies and subgroups within these studies, results are difficult to apply in clinical practice. More research is needed to obtain more insight into the use of these biomarkers, preferably in relation to clinical symptoms, and to identify more direct biomarkers for assessing Zn status.

## 5. Conclusions

ZnD is highly prevalent in healthy 1–3-year-old children in Western Europe. No significant differences were observed in the socio-economic characteristics between children with and without ZnD. Dietary Zn intake was not associated with ZnD. The latter might partially be explained by various dietary factors influencing intestinal Zn absorption. Further studies are needed regarding the influence of diet composition on Zn absorption in young children. Moreover, future research is required to assess reference values for Zn concentrations in relation to clinical symptoms and health outcomes in young children and gain more insight into the use of certain other proposed and promising biomarkers, such as metallothionein.

## Figures and Tables

**Table 1 nutrients-13-03713-t001:** Characteristics and dietary intake of the study population (1–3 years of age) and relation to Zn status.

	All *N = 278*	ZnD *N = 87*	No ZnD *N = 191*	*p*
**Characteristics**				
Age (years)	1.7 (1.2–2.3)	1.6 (1.1–2.3)	1.7 (1.2–2.3)	0.301
Sex (male)	153 (55.0%)	48 (55.2%)	105 (55.0%)	0.975
Ethnicity (Caucasian)	267 (96.0%)	85 (97.7%)	182 (95.3%)	0.511
From Germany	237 (85.3%)	86 (98.9%)	151 (79.1%)	<0.001 *
Parental educational level	(*N* = 231, M = 47)	(*N* = 74, M = 13)	(*N* = 157, M = 34)	
At least one parent with university education	53 (19.1%)	16 (18.4%)	38 (19.4%)	0.743
Neither with university education	178 (64.0%)	58 (66.7%)	120 (62.8%)	
Parental professional status	(*N* = 219, M = 59)	(*N* = 69, M = 18)	(*N* = 150, M = 41)	
At least one parent working	210 (75.5%)	67 (77.0%)	143 (74.9%)	0.723
Neither working	9 (3.2%)	2 (2.3%)	7 (3.7%)	
Daycare attendance	117 (42.1%)	36 (41.4%)	81 (42.4%)	0.817
	(*N* = 276, M = 2)		(*N* = 189, M = 2)	
Weight-for-age-z-score	0.22 (±0.94)	0.20 (±0.99)	0.24 (±0.92)	0.727
	(*N* = 276, M = 2)		(*N* = 189, M = 2)	
Height/length-for-age-z-score	−0.03 (±0.99)	0.04 (±1.04)	−0.07 (±0.97)	0.417
	(*N* = 271, M = 7)		(*N* = 184, M = 7)	
Stunting	3 (1.1%)	1 (1.1%)	2 (1.0%)	*NA*
	(*N* = 271, M = 7)		(*N* = 184, M = 7)	
**Dietary intake**				
Ever breastfed ^§^	177 (63.7%)	64 (73.6%)	113 (59.2%)	0.081
	(*N* = 267, M = 11)		(*N* = 180, M = 11)	
Duration of breastfeeding ^§^				
0–<6 months	102 (57.6%)	34 (53.1%)	68 (60.2%)	0.362
≥6 months	75 (42.4%)	30 (46.9%)	45 (39.8%)	
Ever formula fed ^§^	251 (90.3%)	79 (90.8%)	172 (90.1%)	0.125
	(*N* = 267, M = 11)		(*N* = 180, M = 11)	
Age of introduction of solid foods (months)	(*N* = 273, M = 5)	(*N* = 86, M = 1)	(*N* = 187, M = 4)	
0–6 months	224 (80.6%)	74 (85.1%)	150 (78.5%)	0.243
>6 months	49 (17.6%)	12 (13.8%)	37 (19.4%)	
Main type of milk intake ^†^°	(*N* = 268, M = 10)		(*N* = 181, M = 10)	
Use of primarily cow’s milk	119 (42.8%)	40 (46.0%)	79 (41.4%)	0.719
Use of primarily formula	147 (52.9%)	47 (54.0%)	100 (52.4%)	0.850
Use of dietary supplements	86 (30.9%)	27 (31.0%)	59 (30.9%)	0.798
*(unknown zinc content)*				
Total amount of milk per day ° (mL/day)	420 (400–600)	420 (300–600)	420 (400–600)	0.376
Zn intake from milk in general (mg/day)	2.78 (1.95–3.57)	2.76 (2.05–3.34)	2.79 (1.84–3.68)	0.481
	(*N* = 268, M = 10)		(*N* = 181, M = 10)	
Meat (g/day)	19.3 (11.7–29.7)	19.3 (11.9–29.7)	18.8 (11.0–29.7)	0.611
Zn intake from meat (mg/day)	0.63 (0.38–0.96)	0.63 (0.38–0.96)	0.61 (0.36–0.96)	0.611
	(*N* = 268, M = 10)		(*N* = 181, M = 10)	
Fish (g/day)	2.3 (0.9–4.5)	2.4 (1.2–4.7)	2.0 (0.8–4.4)	0.458
Zn intake from fish (mg/day)	0.02 (0.01–0.03)	0.02 (0.01–0.04)	0.02 (0.01–0.03)	0.458
	(*N* = 268, M = 10)		(*N* = 181, M = 10)	
Vegetables (g/day)	61.0 (35.3–112.4)	63.0 (35.9–132.7)	59.3 (34.6–108.1)	0.369
Zn intake from vegetables (mg/day)	0.57 (0.33–1.05)	0.59 (0.34–1.25)	0.56 (0.32–1.01)	0.369
	(*N* = 268, M = 10)		(*N* = 181, M = 10)	
Dried fruits, seeds and nuts (g/day)	0.6 (0.0–5.8)	0.0 (0.0–3.6)	0.8 (0.0–6.2)	0.068
Zn intake from dried fruits, seeds and nuts (mg/day)	0.01 (0.00–0.12)	0.00 (0.00–0.07)	0.02 (0.00–0.12)	0.068
	(*N* = 268, M = 10)		(*N* = 181, M = 10)	
Bread (g/day)	33.7 (32.4–64.9)	33.7 (32.4–64.9)	46.2 (23.8–64.9)	0.415
Zn intake from bread (mg/day)	0.43 (0.41–0.83)	0.43 (0.41–0.83)	0.59 (0.30–0.83)	0.415
	(*N* = 268, M = 10)		(*N* = 181, M = 10)	
Sandwich spread (g/day)	1.1 (0.0–4.5)	0.9 (0.0–3.5)	1.1 (0.1–4.8)	0.436
Zn intake from sandwich spread (mg/day)	0.01 (0.00–0.03)	0.01 (0.00–0.03)	0.01 (0.00–0.04)	0.436
	(*N* = 267, M = 11)		(*N* = 180, M = 11)	
Breakfast cereals (g/day)	3.0 (0.0–24.1)	5.9 (0.0–32.5)	1.9 (0.0–23.0)	0.230
Zn intake from breakfast cereals (mg/day)	0.06 (0.00–0.47)	0.12 (0.00–0.64)	0.04 (0.00–0.45)	0.230
	(*N* = 268, M = 10)		(*N* = 181, M = 10)	
Eggs (g/day)	6.7 (1.6–6.7)	6.7 (1.6–6.7)	3.9 (1.6–6.7)	0.591
Zn intake from eggs (mg/day)	0.10 (0.02–0.10)	0.10 (0.02–0.10)	0.06 (0.02–0.10)	0.591
	(*N* = 268, M = 10)		(*N* = 181, M = 10)	
Zn intake from solid foods (mg/day)	2.52 (1.86–3.73)	2.76 (2.20–3.73)	2.33 (1.77–3.75)	0.066
	(*N* = 268, M = 10)		(*N* = 181, M = 10)	
Total dietary Zn intake (mg/day)	5.71 (4.43–6.85)	5.59 (4.57–6.74)	5.83 (4.33–6.93)	0.978
*(from milk and solid foods*)	(*N* = 268, M = 10)		(*N* = 181, M = 10)	

Data are expressed as medians (Q1–Q3) or numbers (percentages). In the case of normal distribution, means (standard deviations) are reported. ^§^ Exclusively or partially. ^†^ During the previous month; ° N = 2 other main types of milk/dairy (i.e., human milk (N = 1), yogurt (N = 1)). *p* denotes *p*-value regarding comparisons between “ZnD” and “no ZnD”. * Statistically significant with *p* < 0.05. Abbreviations: Zn, zinc; ZnD, zinc deficiency; N, number; M, missing; NA, not applicable.

## Data Availability

The data presented in this study are available on request from the corresponding author. The data are not publicly available since the participants of this study did not agree for their data to be shared publicly.

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
