# Peer review of "Prevalence of Zinc Deficiency in Healthy 1–3-Year-Old Children from Three Western European Countries"

_nutrients, 2021, doi:10.3390/nu13113713_

Round 1
Reviewer 1 Report
The manuscript describes the zinc status of a sample of 279 children ages 1 to 3 years from 3 different countries in Europe: 85% Germany, 13.6% Netherlands, and 1.4% in the UK.
Introduction:
Line 51: Cohort? – from RCT? (Line 56-57?) cross-sectional?
Methods: Could you please clarify what you mean with “centrally analyzed”?
Laboratory Analysis: What materials were used to collect the blood samples? What precautions to avoid external contamination were taken when blood sample collection? How where the samples collected and stored prior to shipment in each location? Helpful to clarify all these procedures in the manuscript. Was it plasma or serum analyzed?
Where there any sociodemographic, dietary differences between the children included in the sample vs. the 25% that was not included in the analysis?
Line 136. Risk factors:
This is a descriptive observational study using univariate analyses to establish “factors associated with zinc deficiency”. NOT RISK FACTORS. To identify risk factors, a different study design is needed.
Your data is clustered and this may influence your associations. I think it was a good idea to stratify the analysis by age group, however I think it would be even more interesting and a better approach to look at multivariate models (generalized linear mixed models including cluster as random effect) and defining a method to include covariates in the models. (See this reference as an example: https://onlinelibrary.wiley.com/doi/full/10.1111/mcn.12885). You have several borderline significant asociations that may be worth exploring further: “ever breastfeed”. If available, consider exploring “length of exclusive breastfeeding” as a continuous variable in association with zn Deficiency.
Will one non-fasting measure of serum zinc would really be an indicator of zinc deficiency without any additional functional measures? Length/height-for age z-scores (zhaz) are often considered a functional manifestation of zinc deficiency (https://pubmed.ncbi.nlm.nih.gov/17988007/ ) did you have any anthropometric indicators? I think it would be worth to describe zhaz (and stunting if at all present).
I would encourage the authors to adjust all dietary variables by child’s age.
Statistical analysis:
You are collecting samples in three different countries, and the procedures of recruitment and data collection are different. Have you considered including a “cluster” (country) or as random effect when evaluating the factors associated with zinc deficiency?
Lines 143-145 I really like the idea of this sensitivity analysis. Children ages 6 months to 2 years are going through the “complementary feeding period”. Dietary behaviors should be radically different. I would expect to observe more zinc deficiency in younger children. Have the authors consider using a multivariable approach to analyze the factors associated with zinc, and include infant/toddler age as covariate (as well as other important variables that may in theory or by a statistically pre-established rule ? (e.g. Linear mixed or generalized linear mixed models including cluster as random effect, and select variables (dietary)? Was there any information on breastfeeding status and/or milk intake?
Line 135. I would avoid using any type of causal language. Better to use associations as opposed to risk. “factors associated with zinc deficiency”.
Discussion:
Overall, what does it mean to have zinc deficiency with current cutoffs in the described population, without functional symptoms of zinc deficiency?
Like the authors state in the manuscript, cutoffs are based on healthy adult populations that may not be relevant in young children. If there are no functional manifestations of zinc deficiency in this population, (which I highly encourage the authors to further look into e.g. haz if available) and children are healthy, I wonder if it would be worth to further describe and discuss/compare means (sd) of zinc concentrations found in this age group to contribute to the literature on what “normal” zinc concentrations could be in this population and with what is available n the literature… (even stratifying further the data by year of age). There is indeed a gap in the literature on this aspect.
The strengths and limitations of the manuscript are well described.
Author Response
Response to Reviewer 1 Comments
Comments and Suggestions for Authors
The manuscript describes the zinc status of a sample of 279 children ages 1 to 3 years from 3 different countries in Europe: 85% Germany, 13.6% Netherlands, and 1.4% in the UK.
Response:
We thank Reviewer 1 for his/her comments and concerns regarding the manuscript which are one by one addressed below.
Introduction:
Line 51: Cohort? – from RCT? (Line 56-57?) cross-sectional?
Response:
Indeed, this was not a ‘cohort’ based on a longitudinal study in which participants were followed over a period of time, but rather a ‘population’ based on data retrieved from the baseline study visits from the RCT mentioned (i.e. cross-sectional). Introduction, page 2, line 52 of the revised manuscript (without track changes): we removed ‘cohort’ and replaced it with ‘population’.
Methods:
Could you please clarify what you mean with “centrally analyzed”?
Laboratory Analysis: What materials were used to collect the blood samples? What precautions to avoid external contamination were taken when blood sample collection? How where the samples collected and stored prior to shipment in each location? Helpful to clarify all these procedures in the manuscript. Was it plasma or serum analyzed?
Response:
We clarified the procedures regarding the collection and laboratory analyses of blood samples in the revised manuscript. Methods, 2.3. Laboratory Analysis and Definitions, page 3, lines 99-108 of the revised manuscript (without track changes). ‘Venous blood samples were collected from participants throughout various times of the day by trained personnel using trace-element free heparinized tubes, following local laboratory protocol and procedures recommended by the International Zinc Nutrition Consultative Group (IZiNCG). After clotting and centrifugation, serum was distributed and aliquoted in polypropylene tubes and stored between -20 °C and -80 °C at the study site. From the study sites, samples were shipped (within two months) on dry ice to the Nutricia Research Analytical Science Laboratory in Utrecht, the Netherlands, where samples were stored at -80 °C. From the Nutricia Research Analytical Science Laboratory, samples were shipped on dry ice to the Reinier Haga Medical Diagnostic Center in Delft, the Netherlands, where samples were stored at -80 °C until analysis.’ Moreover, we clarified that serum (not plasma) Zn concentrations were analyzed. Methods, 2.3. Laboratory Analysis and Definitions, page 3, line 108-111 of the revised manuscript. ‘Serum Zn concentrations were determined using flame atomic absorption spectrophotometry (AA-7000, Shimadzu), and hsCRP concentrations using a clinical chemistry analyzer (Abbott Architect c16000) with a turbidimetry method.’
Where there any sociodemographic, dietary differences between the children included in the sample vs. the 25% that was not included in the analysis?
Response:
There were no statistically significant socio-economic or dietary differences between the children who were included and those who were excluded. We added this information in the manuscript, Results, 3.1. Study Population, page 3, lines 135-137 of the revised manuscript (without track changes): ‘There were no statistically significant differences regarding socio-economic characteristics and dietary intake found between included and excluded children (data not shown).’
Line 136. Risk factors:
This is a descriptive observational study using univariate analyses to establish “factors associated with zinc deficiency”. NOT RISK FACTORS. To identify risk factors, a different study design is needed.
Response:
We agree that based on the type of our study and the analyses performed one can only speak of associated factors rather than risk factors. We replaced ‘risk factors for ZnD’ for ‘factors associated with ZnD’ in Results, 3.3. Factors Associated with ZnD, page 5, line 160 of the revised manuscript (without track changes), and throughout the manuscript.
Your data is clustered and this may influence your associations. I think it was a good idea to stratify the analysis by age group, however I think it would be even more interesting and a better approach to look at multivariate models (generalized linear mixed models including cluster as random effect) and defining a method to include covariates in the models. (See this reference as an example: https://onlinelibrary.wiley.com/doi/full/10.1111/mcn.12885). You have several borderline significant associations that may be worth exploring further: “ever breastfeed”.
Response:
Besides stratifying the analyzed population by age group, we performed a multivariate analysis as recommended. This information was added to the Methods and Results section:
Methods, 2.4. Statistical Analysis, page 3, lines 126-129 of the revised manuscript (without track changes) ‘In order to further explore factors associated with ZnD, a binary logistic regression analysis was performed using the variables previously identified with a p-value <0.1 as covariates, including age and sex independent of p-values, and ZnD as dependent variable.’ Results, 3.3. Factors Associated with ZnD, page 5, lines 170-178 of the revised manuscript (without track changes). ‘A binary logistic regression analysis was performed to further explore factors associated with ZnD, with ‘ZnD’ as dependent variable and ‘age’, ‘sex’, ‘ever breastfed’, ‘from Germany’, and ‘dietary Zn intake’ as covariates, and showed that being ‘from Germany’ was significantly associated with ZnD (OR=16.6; 95% CI 92.2-124.2; p=0.006). No other significant associations with ZnD were found (data not shown). Although ‘Zn intake from dried fruits, seeds and nuts’ had a p-value <0.1, we chose not to include this variable in our multivariate model since this variable is already included in ‘dietary Zn intake’. A separate binary logistic analysis with ‘Zn intake from dried fruits, seeds and nuts’ instead of ‘dietary Zn intake’, showed no significant association with ZnD (data not shown).
If available, consider exploring “length of exclusive breastfeeding” as a continuous variable in association with zn Deficiency.
Response:
We agree that the ‘length of exclusive breastfeeding’ as a continuous variable in association with ZnD would be interesting to explore since children with (prolonged) exclusive breastfeeding and delayed introduction of complementary foods are at risk for developing ZnD. Unfortunately, the ‘length of exclusive breastfeeding’ as a continuous variable is not available to us, and therefore we were not able to explore this association further.
Will one non-fasting measure of serum zinc would really be an indicator of zinc deficiency without any additional functional measures? Length/height-for age z-scores (zhaz) are often considered a functional manifestation of zinc deficiency (https://pubmed.ncbi.nlm.nih.gov/17988007/ ) did you have any anthropometric indicators? I think it would be worth to describe zhaz (and stunting if at all present).
Response:
Anthropometric z-scores (i.e. weight-for-age-z-scores and height/length-for-age-z-scores) were added in Table 1 (see Table 1.). Percentages of stunting was added in Table 1 (see Table 1.) and described in Results, 3.1. Study Population, page 3, lines 139-140 of the revised manuscript (without track changes): ‘Stunting was present in three children (1.1%)’. Due to the small number of stunting,, no additional statistical analysis was performed regarding the association with ZnD (see Table 1.). Moreover, additional information regarding the collection of anthropometric data and the definition of stunting was provided in the Methods section. Methods, 2.2. Study Population and Procedure, page 2, lines 81-88 of the revised manuscript (without track changes).
I would encourage the authors to adjust all dietary variables by child’s age.
Response:
A subgroup analysis showed that ZnD-prevalence did not significantly differ between 1-2-year old and 2-3-year old children, despite possible differences in dietary intake details. Moreover, the additional binary logistic regression analysis showed that age was not associated with ZnD. The aim of this study was to assess ZnD-prevalence and associated factors, and age was not associated with ZnD. We understand that it might be interesting for the reader to have some insight in (dietary) variables in 1-2-year old vs. 2-3-year old children. However, since it does not directly address the aim of this study, we believe it might be distracting or even confusing for some readers to provide this information. Therefore, we chose not to present this information in the manuscript.
Statistical analysis:
You are collecting samples in three different countries, and the procedures of recruitment and data collection are different. Have you considered including a “cluster” (country) or as random effect when evaluating the factors associated with zinc deficiency?
Response:
As previously mentioned, we performed a multivariate analysis as recommended and this information was added to the Methods section: Methods, 2.4. Statistical analysis, page 3, lines 126-129 of the revised manuscript (without track changes):‘In order to further explore factors associated with ZnD, a binary logistic regression analysis was performed using the variables previously identified with a p-value <0.1 as covariates, including age and sex independent of p-values, and ZnD as dependent variable.’, as well as in the Results: Results, 3.3. Factors Associated with ZnD,, page 5, lines 170-178 of the revised manuscript (without track changes): ‘A binary logistic regression analysis was performed to further explore factors associated with ZnD , with ‘ZnD’ as dependent variable and ‘age’, ‘sex’, ‘ever breastfed’, ‘from Germany’, and ‘dietary Zn intake’ as covariates, and showed that being ‘from Germany’ was significantly associated with ZnD (OR=16.6; 95% CI 92.2-124.2; p=0.006). No other significant associations with ZnD were found (data not shown). Although ‘Zn intake from dried fruits, seeds and nuts’ had a p-value <0.1, we chose not to include this variable in our multivariate model since this variable is already included in ‘dietary Zn intake’. A separate binary logistic analysis with ‘Zn intake from dried fruits, seeds and nuts’ instead of ‘dietary Zn intake’, showed no significant association with ZnD (data not shown)’. This binary logistic regression analysis showed that being from Germany was associated with 16.6 times increased odds of ZnD. We cannot rule out that this association is possibly due to the difference in setting and recruitment procedures (e.g. fasting state) between children recruited from Germany and children recruited from the Netherlands and the UK. However, we believe that since the majority of participants were from Germany (85%), it is likely by chance that children with ZnD are more likely to come from Germany than from the Netherlands or the UK. This was included in the Discussion, 4.2. Factors Associated with ZnD, page 6, lines 205-211 of the revised manuscript (without track changes).
Lines 143-145 I really like the idea of this sensitivity analysis. Children ages 6 months to 2 years are going through the “complementary feeding period”. Dietary behaviors should be radically different. I would expect to observe more zinc deficiency in younger children. Have the authors consider using a multivariable approach to analyze the factors associated with zinc, and include infant/toddler age as covariate (as well as other important variables that may in theory or by a statistically pre-established rule ? (e.g. Linear mixed or generalized linear mixed models including cluster as random effect, and select variables (dietary)?
Response:
ZnD-prevalence rates did not significantly differ between 1-2-year old and 2-3-year old children, despite possible differences in dietary behaviors. Moreover, multivariate analysis showed that age was not associated with ZnD, and did not reveal a higher presence of ZnD in younger children. Furthermore, the age of introduction of solid foods/complementary feeding (albeit dichotomous) did not differ between children with and without ZnD.
Was there any information on breastfeeding status and/or milk intake?
Response:
The ‘information on breastfeeding status and/or milk intake’ that is available (ever breastfed (i.e. exclusive or partially); duration of breastfeeding (i.e. exclusive or partially), dichotomous (0-6 months, ≥ 6 months); ever formula fed (i.e. exclusive or partially); main type of milk intake (either primarily cow’s milk or formula); total amount of milk per day (mL/day); and Zn intake from milk per day (mg/day)) is provided in Table 1. Regarding ‘breastfeeding status’, as previously mentioned, information on the ‘length of exclusive breastfeeding’ is unfortunately not available.
Line 135. I would avoid using any type of causal language. Better to use associations as opposed to risk. “factors associated with zinc deficiency”.
Response:
We replaced ‘risk factors for ZnD’ for ‘factors associated with ZnD’ in Results, 3.3. Factors Associated with ZnD,, page 5, lines 160 of the revised manuscript (without track changes), and throughout the manuscript, as previously mentioned.
Discussion:
Overall, what does it mean to have zinc deficiency with current cutoffs in the described population, without functional symptoms of zinc deficiency?
Response:
This is a legitimate question/concern that we address in the Discussion. Besides the debate on whether or not Zn concentrations (accurately/adequately) reflect zinc status, it is indeed questionable what it means to have ZnD based on current used cut-offs if functional/clinical symptoms are absent. As stated in the Discussion it is currently unknown whether Zn concentrations below a certain predefined cut-off value (e.g. based on cohort-derived reference values), fail to meet bodily requirements and causes negative health consequences. In adults, a lower cut-off value of 50 µg/dL (i.e. 7.6 µmol/L) has been proposed for clinical ZnD associated with symptoms (Wessels et al. 2014). However, to our knowledge, no such studies were/are being conducted in (young) children. In our opinion, it would be interesting to investigate cut-off values for Zn concentrations in relation to symptoms to address this question/concern. In our study, besides anthropometric z-scores and % of stunting, no other (functional) health outcomes were available to us to relate to ZnD. In our study, stunting was only prevalent in three children (1.1%). However, due to this small number no further statistical analysis could be performed regarding a possible association with ZnD.
Like the authors state in the manuscript, cutoffs are based on healthy adult populations that may not be relevant in young children. If there are no functional manifestations of zinc deficiency in this population, (which I highly encourage the authors to further look into e.g. haz if available) and children are healthy, I wonder if it would be worth to further describe and discuss/compare means (sd) of zinc concentrations found in this age group to contribute to the literature on what “normal” zinc concentrations could be in this population and with what is available in the literature… (even stratifying further the data by year of age). There is indeed a gap in the literature on this aspect.
Response:
As stated in the Discussion, an important limitation of this study is that we did not have information concerning the fasting state during blood draw, the time of blood draw and meal consumption prior to the blood draw, which are factors known to possibly influence Zn concentrations. In order to establish “normal” zinc concentrations in the youngest age groups, we think that these factors need to be included in (future) studies. Moreover, as previously described, even if “normal” zinc concentrations (including above mentioned factors known to possibly influence zinc concentrations) are being researched, the question remains what these cut-offs mean if not related to (functional) health outcomes of ZnD. With respect to abovementioned concerns, we believe this study was not appropriate for mentioning “normal” zinc concentrations in this population.
The strengths and limitations of the manuscript are well described.
Reviewer 2 Report
The study is interesting since it shows a negative condition in relationship to zinc deficiency in 1-3 old age children of Western Europe. However, I think in the introduction it is necessary to better define the consequences of zinc deficiency and, in the discussion, to give appropriate news on a possible therapy of this condition. In addition, the authors declare that the evaluation of zinc can be obtained by other than plasma/sera tissues. For this reason, the authors have to explain the choice to evaluate zinc values only in peripheral blood.
Author Response
Response to Reviewer 2 Comments
Comments and Suggestions for Authors:
The study is interesting since it shows a negative condition in relationship to zinc deficiency in 1-3 old age children of Western Europe. However, I think in the introduction it is necessary to better define the consequences of zinc deficiency (1) and, in the discussion, to give appropriate news on a possible therapy of this condition (2). In addition, the authors declare that the evaluation of zinc can be obtained by other than plasma/sera tissues. For this reason, the authors have to explain the choice to evaluate zinc values only in peripheral blood (3).
Response:
We thank Reviewer 2 for his/her comments and concerns regarding the manuscript. We have addressed these concerns below.
1.) Since zinc is involved in numerous processes affecting all kinds of cell types and tissues throughout the body; for example in the skin, zinc is found in both dermis and epidermis, resulting in a variety of possible skin lesion/disorders in case of ZnD. We tried to give an overview of the (wide range of) possible effects of ZnD, but we might have been to general in doing so. We better defined the consequences of ZnD mentioned in the Introduction, page 1, line 35-38 of the revised manuscript (without track changes): ‘…including growth retardation, skin lesions and delayed wound healing, and impaired innate and adaptive immune functions.’. We removed ‘neurobehavioral’ due to the lengthiness of the sentence. Furthermore, more relevant references were added.
2.) The aim of this study was to assess the prevalence of ZnD in a population of 1-3-year-old children, and to evaluate possible associated factors with ZnD. We did not evaluate (possible or appropriate) treatment/therapy for ZnD, and therefore we feel that we cannot conclude or make a statement as to what treatment/therapy for ZnD (in this population) is appropriate. As addressed in the Discussion, it is debatable what it means to have ZnD based on (current) used cut-off values, especially when (functional) adverse health outcomes /clinical symptoms of ZnD are absent. Is ZnD present when (functional) adverse health outcomes /clinical symptoms are absent? And more importantly should we treat ZnD (e.g. start Zn supplementation or provide dietary recommendations?) based on (only) numbers? It is currently unknown whether Zn concentrations below a certain predefined cut-off value (e.g. based on cohort-derived reference values), fail to meet bodily requirements and causes negative health consequences. In adults, a lower cut-off value of 50 µg/dL (i.e. 7.6 µmol/L) has been proposed for clinical ZnD associated with symptoms (Wessels et al. 2014). However, to our knowledge, no such studies were/are being conducted in (young) children. In our opinion, it would be interesting to investigate cut-off values for Zn concentrations in relation to symptoms to address this concern. In our study, besides anthropometric z-scores and % of stunting, no other (functional) health outcomes were available to us to relate to ZnD. Overall, stunting was only prevalent in three children (1.1%). However, due to this small number, no additional statistical analysis could be performed regarding a possible association with ZnD. Furthermore, with respect to dietary recommendations as ‘therapy/treatment’ for ZnD, it is questionable as to what recommendations are adequate, if at all effective. In our study, we found no association between dietary Zn intake (in total, but also by reported food group) and ZnD. As stated in the Discussion, the lack of an association between dietary Zn intake and Zn concentration might be explained by factors that influence intestinal Zn absorption (such as phytates and the amount of dietary protein). In short, the composition of the diet might impact Zn absorption considerably, despite sufficient Zn intake. Our study does not (sufficiently) provide insight on this matter, and we therefore opt for further research regarding the influence of diet composition on Zn absorption in young children.
3.) This study was a cross-sectional analysis that used baseline data from a previously conducted randomized, double-blind controlled trial investigating the effect of a micronutrient-fortified growing up milk given for 20 weeks on the iron and vitamin D status of healthy 1-3-year-old children in comparison to non-fortified cow’s milk. This effect was the primary objective of the RCT, and Zn concentrations were described as one of the exploratory outcome parameters in the study protocol. Due to the ‘burden’ of the intervention already, only (extra) blood was collected during the (baseline) study visit, and not for example urine and hair as well. Moreover, as described in the Discussion, Zn concentration is the most extensively investigated and widely used biomarker to assess zinc status. In short, at this point, it is the best we’ve got.
Reviewer 3 Report
1. The title is appropriate for the first part. This study is still considered a small study (at least not a population- based study). It is not sure whether "from three Western European countries" is appropriately used for the small scale study.
2. The aim of this study is appropriately described. The risk factors that are described to be analyzed for the association between ZnD and No ZnD is misleading. It basically conducted the comparison between these two groups in terms of the difference for the variables of interest. There is no further statistical data analysis to be conducted where underlying factor(s) may be actually a risk factor when adjusted for the other variables.
3. The study participants are from three settings (or three countries-- Germany, Netherlands and UK). Participants in the Germany were from outpatient pediatric clinics, and those from Netherlands were from hospital setting that were for pre-operative visits or non-emergency procedures. They are different participants with different health status (and this reviewer disagrees with the authors that those scheduled for pre-operative visits were considered "healthy"). These study subjects were not considered homogenous and they should not be combined together for the data analysis. At least, some children are not considered "healthy" to this reviewer.
4. The study participants aged from 1 year-old to 3-year-old.. Children aged 2.5 vs. aged 1.2, for example, have quite a difference in neuro-development and energy intake and activity levels. The characteristics included did not have total calorie intake per day information (or adjusted calorie intake per Kilogram/day, e.g.,). Some of the variables included, such as dried fruits, seeds, and nuts per day have quite a range of variation. Some variables had high missing numbers (e.g., parental education levels and professional status). The above conditions probably resulted from the inconsistent data collection from the three different study sites (or countries).
5. The paper still needs further refinement and be precise on the description of the data that are presented (and the results that are shown). The abstract also needs modification.
Author Response
Response to Reviewer 3 Comments
Comments and Suggestions for Authors
Response:
We thank Reviewer 2 for his/her comments and concerns regarding the manuscript which are addressed below.
- The title is appropriate for the first part. This study is still considered a small study (at least not a population- based study). It is not sure whether "from three Western European countries" is appropriately used for the small-scale study.
Response:
We do acknowledge that it may seem a bit far-fetched to state “from three Western European countries”, since only 37 participants (13.3%) were from the Netherlands and four (1.4%) from the UK, and the majority (n=237, 85.3%) from Germany. However, participants were recruited in (these) three European countries. Moreover, since only few studies report on ZnD-prevalence rates in young children from Western (European) high-income countries (that were conducted some time ago, had relatively small study populations and did not extensively assess associated factors), we believe mentioning “from three Western European countries” should be mentioned in the title. We removed ‘large’ in the Introduction, page 2, line 52 of the revised manuscript (without track changes) and as well as ‘population-based’ in the Introduction, page 2, line 50 of the revised manuscript (without track changes). In the Discussion, 4.3. Strengths and Limitations, page 6, line 228 of the revised manuscript (without track changes) ‘relatively’ was added and ‘homogenous’ removed: ‘Strengths of this study are the relatively large and homogenous study population…’. ‘Relatively’ in comparison to previously mentioned studies reporting on ZnD-prevalence rates in young children from Western (European) high-income countries.
- The aim of this study is appropriately described. The risk factors that are described to be analyzed for the association between ZnD and No ZnD is misleading. It basically conducted the comparison between these two groups in terms of the difference for the variables of interest. There is no further statistical data analysis to be conducted where underlying factor(s) may be actually a risk factor when adjusted for the other variables.
Response:
We agree with Reviewer 3 that based on the analyses performed we can only speak of associations rather than risk factors. We replaced ‘risk factors for ZnD’ for ‘factors associated with ZnD’ throughout the manuscript. Furthermore, we now also included a multivariate analysis to adjust for possible confounders, see response below for more details.
- The study participants are from three settings (or three countries-- Germany, Netherlands and UK). Participants in the Germany were from outpatient pediatric clinics, and those from Netherlands were from hospital setting that were for pre-operative visits or non-emergency procedures. They are different participants with different health status (and this reviewer disagrees with the authors that those scheduled for pre-operative visits were considered "healthy"). These study subjects were not considered homogenous and they should not be combined together for the data analysis. At least, some children are not considered "healthy" to this reviewer.
Response:
We clarified ‘healthy’ in the Methods section; Methods, 2.2. Study Population and Procedure, page 2, lines 69-72 of the revised manuscript (without track changes): ‘Children between 1-3 years of age with stable health status (i.e. being unknown with chronic or recent acute diseases, no known infection during the last week or infection needing medical assistance or treatment during the last two weeks) and expected to remain stable were eligible for participation’. Moreover, children with CRP >5 mg/L were excluded from analysis. In this respect, we consider these children ‘healthy’. Furthermore, the majority of the elective non-emergency surgical procedures were general surgery/urological procedures, for example correction of inguinal hernia and orchidopexia, in otherwise healthy children.
We performed an additional multivariate analysis, Results, 3.3. Factors Associated with ZnD,, page 5, lines 170-178 of the revised manuscript (without track changes). ‘A binary logistic regression analysis was performed to further explore factors associated with ZnD, with ‘ZnD’ as dependent variable and ‘age’, ‘sex’, ‘ever breastfed’, ‘from Germany’, and ‘dietary Zn intake’ as covariates, and showed that being ‘from Germany’ was significantly associated with ZnD (OR=16.6; 95% CI 92.2-124.2; p=0.006). No other significant associations with ZnD were found (data not shown). Although ‘Zn intake from dried fruits, seeds and nuts’ had a p-value <0.1, we chose not to include this variable in our multivariate model since this variable is already included in ‘dietary Zn intake’. A separate binary logistic analysis with ‘Zn intake from dried fruits, seeds and nuts’ instead of ‘dietary Zn intake’, showed no significant association with ZnD (data not shown)’. This binary logistic regression analysis showed that being from Germany was associated with 16.6 times increased odds of ZnD. We cannot rule out that this association is possibly due to the difference in setting and recruitment procedures (e.g. fasting state) between children recruited from Germany and children recruited from the Netherlands and the UK. However, we believe that since the majority of participants were from Germany (85%), it is likely by chance that children with ZnD are more likely to come from Germany than from the Netherlands or the UK. This was included in the Discussion, 4.2. Factors Associated with ZnD, page 6, lines 207-211 of the revised manuscript (without track changes).
- The study participants aged from 1 year-old to 3-year-old. Children aged 2.5 vs. aged 1.2, for example, have quite a difference in neuro-development and energy intake and activity levels. The characteristics included did not have total calorie intake per day information (or adjusted calorie intake per Kilogram/day, e.g.,).
Response:
We do not have data regarding neuro-development, energy intake (total calorie intake per day/ adjusted calorie intake per kg/day) or activity levels of our study participants. This study showed no association between dietary Zn intake (in total, but also by reported food group) and ZnD. As stated in the Discussion, the lack of an association between dietary Zn intake and Zn concentration might be explained by factors that influence intestinal Zn absorption (such as phytates and the amount of dietary protein). The diet composition might impact Zn absorption, despite sufficient Zn intake. Further research regarding the influence of diet composition on Zn absorption in young children is needed, in which factors such as energy intake (calorie intake per (kg/)day) and activity levels might be interesting to investigate.
Some of the variables included, such as dried fruits, seeds, and nuts per day have quite a range of variation. Some variables had high missing numbers (e.g., parental education levels and professional status). The above conditions probably resulted from the inconsistent data collection from the three different study sites (or countries).
Response:
Indeed, the conditions mentioned might have resulted from the inconsistent data collection from the different study sites/countries.
- The paper still needs further refinement and be precise on the description of the data that are presented (and the results that are shown). The abstract also needs modification.
Response:
We hope that the modifications and additions in the revised manuscript, and our explanation(s) were able to address your concerns accordingly. The abstract was modified.
Round 2
Reviewer 3 Report
No further comments